# Misleading Positive Serology for Cat Scratch Disease following Administration of Intravenous Immunoglobulin

**DOI:** 10.3390/pathogens11020177

**Published:** 2022-01-27

**Authors:** Michal Yakubovsky, Yoav Golan, Alex Guri, Itzhak Levy, Daniel Glikman, Moshe Ephros, Michael Giladi

**Affiliations:** 1Infectious Disease Unit, Tel Aviv Sourasky Medical Center, Tel Aviv University, Tel Aviv 6423906, Israel; michalya@tlvmc.gov.il; 2The Sackler Faculty of Medicine, Tel Aviv University, Tel Aviv 6997801, Israel; 3Tufts Medical Center, The Tufts University School of Medicine, Boston, MA 02446, USA; ygolan@tuftsmedicalcenter.org; 4Department of Pediatrics, Kaplan Medical Center, Rehovot 76100, Israel; Alexgur@clalit.org.il; 5The School of Medicine, The Hebrew University and Hadassah Medical Center, Jerusalem 7661041, Israel; 6Pediatric Infectious Disease Unit, Schneider Children’s Medical Center, Petah Tiqva 4920235, Israel; itzhakl@clalit.org.il; 7Infectious Disease Unit, Padeh Poriya Medical Center, Poriya 1528001, Israel; daniel.glikman@biu.ac.il; 8The Azrieli Faculty of Medicine in the Galilee, Bar-Ilan University, Safed 1311502, Israel; 9Pediatric Infectious Disease Unit, Carmel Medical Center, Haifa 3436212, Israel; mefrat@technion.ac.il; 10The Rappaport Faculty of Medicine, Technion-Israel Institute of Technology, Haifa 3200003, Israel; 11The Bernard Pridan Laboratory for Molecular Biology of Infectious Diseases, Tel Aviv Sourasky Medical Center, Tel Aviv 6423906, Israel

**Keywords:** cat scratch disease, *Bartonella henselae*, serology, enzyme immunoassay, intravenous immunoglobulin, false positive results

## Abstract

Cat scratch disease (CSD), caused by *Bartonella henselae*, usually presents as regional lymphadenopathy/lymphadenitis, known as typical CSD or as atypical CSD, which includes, among others, neurological manifestations. Serology for anti-*B. henselae* IgG antibodies is the most commonly used diagnostic tests for CSD. Intravenous immunoglobulin (IVIG) is given for an increasing number of medical conditions and may cause interference with serological testing. We report six patients with neurological manifestations and two patients with Kawasaki disease mimicking typical CSD, mistakenly diagnosed as CSD due to false-positive serology following IVIG therapy. *Bartonella* IgG serology was positive one to six days after IVIG administration and reverted to negative in seven of eight patients or significantly decreased (1 patient) ≤30 days later. In patients with CSD, IgG titers remained essentially unchanged 15–78 days after the positive serum sample. An additional eight patients treated with IVIG for various conditions were evaluated prospectively. All were seronegative one day pre-IVIG infusion, five patients demonstrated an increase in the IgG titers one to three days after IVIG administration, one interpreted as positive and four as intermediate, whereas three patients remained seronegative, suggesting that false seropositivity after IVIG therapy may not occur in all patients. Treatment with IVIG can result in false-positive serology for *B. henselae.* Increased awareness to the misleading impact of IVIG is warranted to avoid misinterpretation. Repeat testing can distinguish between true and false serology. Preserving serum samples prior to IVIG administration is suggested.

## 1. Introduction

Cat scratch disease (CSD) is caused mostly by *Bartonella henselae*, a fastidious, gram negative bacillus found worldwide in bacteremic cats [1,2]. Typical CSD, characterized by self-limited regional lymphadenopathy or lymphadenitis, often associated with constitutional symptoms such as fever, malaise, and night sweats, occurs in 85–90% of CSD patients. Approximately 10% of CSD patients have extra nodal manifestations collectively known as atypical CSD and include, among others, musculoskeletal, ophthalmological, hepato-splenic, dermatological, and neurological manifestations. Among the latter, encephalitis and meningoencephalitis are the most common, with transverse myelitis and Guillain-Barré syndrome reported rarely [2,3,4,5,6,7]. Laboratory diagnosis of CSD is still problematic owing to the limitations of available confirmatory tests. *B. henselae* culture from affected lymph nodes is rarely positive, and growth, if occurs, may require up to four weeks of incubation; Warthin Starry silver stain is of low sensitivity and inadequate specificity; cytology and histopathology are not specific; immunohistochemical assay is not available in routine diagnostic laboratories, and polymerase chain reaction (PCR), although sensitive and specific, requires tissue or pus specimens from involved tissue, which are not commonly available [3]. Serological assays, both immunofluorescent antibody (IFA) test and enzyme immunoassay (EIA), for detection of anti-*B. henselae* antibodies have become the most commonly used diagnostic tests for CSD, both as in-house and commercial products. One of the several limitations of CSD serology is the frequent lack of anti-*Bartonella* IgM, even in documented acute infection. Hence, the majority of CSD cases are diagnosed based on the presence of anti-*B. henselae* IgG [3,8].

Intravenous immunoglobulin (IVIG) has become a relatively common treatment for immune-mediated and inflammatory diseases and primary or secondary immunodeficiency states [9]. Interference with serological testing of IgG antibodies is a recognized, though not well characterized, phenomenon after administration of immunoglobulin and there are a number of reports describing false-positive serological testing with potential misleading diagnoses of infectious disease following IVIG administration [10,11,12,13,14,15,16]. In the present study, we report for the first time a mistaken diagnosis of typical and atypical CSD in patients with clinical presentations consistent with CSD due to false-positive serological testing following therapy with IVIG and present an approach to avoid an incorrect diagnosis.

## 2. Results

The following representative cases illustrate the clinical quandary about serodiagnosis of CSD following treatment with IVIG.

Case 1, IVIG given for encephalitis. A previously healthy eight-year-old female was hospitalized because of fever and cough of two weeks’ duration with no response to oral amoxicillin. The patient owned a kitten with whom she used to play frequently. On admission she was febrile and tachypneic, chest x-ray revealed unilateral pneumonia and intravenous ceftriaxone was started. On the third day of admission, her temperature increased to 39.7 °C and she became sleepy and irritable with nuchal rigidity. She later became comatose, was intubated, and transferred to the pediatric intensive care unit. Blood leukocyte count was 27,600/mm^3^, cerebrospinal fluid showed a leukocyte count of 138/mm^3^ with 66% mononuclear cells, protein 75 mg/dL and normal glucose level. Blood and CSF cultures were sterile. Magnetic resonance imaging revealed an abnormal signal in the thalamus, basal ganglia, pons, and the fourth ventricle periventricular white matter, consistent with encephalitis. Empiric treatment included high dose ceftriaxone, acyclovir, dexamethasone, and IVIG 2 gr/kg, divided on two consecutive days. A workup for the diagnosis of a host of viral, bacterial, and fungal pathogens was negative except for positive IgM and IgG serology for *Mycoplasma pneumoniae* and positive anti-*Bartonella* IgG antibodies at a titer of 1:100. Azithromycin and doxycycline were started upon receiving the positive serological results. Serology was repeated 14 days later, demonstrating no change in the IgM and IgG seropositivity to *Mycoplasma pneumoniae* and negative results for *B. henselae*. The patients regained consciousness, but remained with severe neurological deficits. She was transferred to a rehabilitation center after four weeks of hospitalization.

Case 2, IVIG given for atypical Kawasaki disease. A previously healthy four-year-old male was admitted to the pediatric department for investigation of fever of seven days’ duration, maximal temperature 39.5 °C, unilateral cervical lymphadenopathy measuring 6–8 cm, non-specific, generalized maculopapular rash, and bilateral bulbar conjunctival injection. The patient owned a kitten and was often scratched by it. Abdominal ultrasound showed mild splenomegaly. Echocardiogram was within normal limits. An ophthalmologic examination revealed bilateral anterior uveitis. The main differential diagnosis included two entities: (i) atypical Kawasaki disease based on clinical criteria (fever, rash, lymphadenopathy, and conjunctivitis) and characteristic laboratory results; sedimentation rate of >120 mm, C-reactive protein 124 mg/dL (normal ≤5 mg/dL), and hemoglobin 9.3 g/dL. (ii) Cat scratch disease, based on close cat contact, fever, and regional lymphadenopathy. To reduce the risk of Kawasaki disease-associated coronary artery abnormalities, IVIG 2 gr/kg in two divided doses was administered. To rule out concurrent infections, serologic workup for viral and bacterial infections was carried out with only anti-*B. henselae* IgG being positive at a titer of 1:100, thus meeting the diagnostic criteria of CSD. Doxycycline was commenced for seven days. A second serum specimen for *B. henselae* serology was repeated eight days after the first one and was found negative. After 14 days of admission, the patient was discharged with significant clinical and laboratory improvement. On follow up visits, he continued to improve without evidence of cardiac involvement.

Table 1 presents eight patients (including cases 1 and 2 presented above) treated with IVIG produced by various manufacturers given at a dose of 2 gr/kg in two divided doses; six patients with neurological conditions, including encephalitis, meningoencephalitis, transverse myelitis, and Guillain-Barré syndrome and 2 patients with atypical Kawasaki disease. These eight patients were suspected to have CSD based on an overlapping clinical presentation consistent with this diagnosis, which was “corroborated” by positive anti-*B. henselae* IgG serology following treatment with IVIG. In all cases, *Bartonella* IgG serology was positive (≥1:100) within one to six days after IVIG administration and in seven of eight patients reverted to negative serology within 7–14 days to one month after IVIG therapy. In one patient with Guillain-Barré syndrome, the first serum specimen taken two days after IVIG treatment tested positive for anti-*B. henselae* IgG at a titer of 1:200, which declined to 1:100 and <1:100 (interpreted as intermediate) 7 and 15 days after treatment, respectively.

The kinetics of anti-*B. henselae* IgG antibodies in two representative patients with false-positive serology, one with encephalitis and one with atypical Kawasaki disease, demonstrate that IgG serology reverted to negative within ≤30 days (Figure 1). The control group consisted of 24 CSD patients, each with two serum samples taken a mean of 34.9 ± 15.6 days apart. The mean anti-*B. henselae* titer of the first and second serum samples was 86 ± 26 EU and 100 ± 24 EU, respectively, with all second serum samples being seropositive (Figure 1).

Eight patients were evaluated prospectively for the presence of anti-*B. henselae* IgG antibodies prior to and following treatment with IVIG (Table 2). All patients were seronegative one day prior to IVIG administration. Five patients demonstrated an increase in the IgG titers one to three days after IVIG administration interpreted as positive in one patient and intermediate in four patients, whereas three patients remained seronegative after IVIG.

## 3. Discussion

In the current report, we describe eight pediatric patients with clinical presentations consistent with neurological complications of CSD or typical CSD prompting their physicians to corroborate the diagnosis by serological testing. Encephalitis, a predominantly clinical diagnosis, is the most commonly recognized neurologic manifestation of CSD. Among patients with an identified etiology of encephalitis, *B. henselae* is the most common bacterial cause. Nearly half of patients develop seizures, often with status epilepticus, as occurred in two children with encephalitis described herein. Transverse myelitis and Guillain-Barré syndrome have been described in CSD [3,4,5,6,7]. History of close cat or kitten contact, as was reported in three of our patients, added a strong epidemiological support for CSD diagnosis. The positive result of *Bartonella* antibody testing, subsequently proved to be false positive, provided the “evidence” for a laboratory-confirmed diagnosis, in accordance with our CSD case definition and prompted treatment with doxycycline or azithromycin in most of these patients. Two patients were diagnosed with atypical Kawasaki disease, an acute febrile vasculitis of childhood, which can manifest with unilateral cervical lymphadenitis. Febrile illness with unilateral cervical lymphadenitis in the absence of pharyngitis or tonsillitis, though not specific, is also highly suggestive of typical CSD, particularly after obtaining a history of cat exposure, and warrants serologic testing for *Bartonella*. The other manifestations described in these patients, including conjunctivitis, uveitis, rash, and arthralgia, have all been described in CSD as well as in Kawasaki disease [3,17,18].

Diagnosis of CSD was confirmed serologically by EIA, previously shown to be highly specific for the diagnosis of CSD. Specificity was determined by testing patients with non-CSD definite diagnoses, including 144 patients with infectious and 53 patients with non-infectious etiologies and was found to be ≥95% [19]. Another previous study from our laboratory demonstrated that anti-*B. henselae* IgG titers remain high for two to three years after disease onset and although the level of IgG antibodies decreases over time, the rate of the decay is usually too slow to allow detection of a significant decrease over intervals of two to four weeks [20]. Our findings regarding 24 CSD patients tested serologically a mean of 35 days apart and remained seropositive on the second serum sample, also confirm this conclusion. These results are in accordance with the observations of previous reports using EIA or IFA test that also showed a slow decrease during 12 months with lack of significant change in anti-*B. henselae* IgG titers tested at six-week or 100-day intervals [21,22,23]. In contrast, IgG mean half-life of IVIG products ranges from three to five weeks, with big variability among studies, and leads to a rapid IgG decrease following IVIG administration as a function of immunoglobulin half-life [9,24,25,26]. The rapid seroconversion from positive to negative in our patients, occurring within 30 days, strongly argues that the initial high anti-*B. henselae* IgG titers in our patients were false positive, most likely due to the effect of IVIG therapy. Bartonella IFA sensitivity and specificity vary substantially in different laboratories in the US and Europe and it is difficult to predict the effect of IVIG administration on anti-*B. henselae* IgG titers if tested by an IFA assay.

With varying degrees of proven benefit, IVIG has been used as part of the treatment for the medical conditions clinically diagnosed in our patients. IVIG is an FDA-approved indication for prevention of coronary artery aneurysms in Kawasaki disease, is considered definitely beneficial in Guillain-Barré syndrome, and is occasionally used as an adjunctive empiric therapy in children with encephalitis or patients with transverse myelitis of undetermined etiology. This approach is supported by empirical evidence of an IVIG-associated beneficial response in the treatment of viral and autoimmune encephalitis and the assumption that irrespective of etiology, the underlying pathogenesis in encephalitis and myelitis is brain or spinal cord inflammation that may be ameliorated by IVIG [9,27].

Passive acquisition of various clinically important antibodies through therapies with IVIG has been reported before, particularly regarding hepatitis B surface and core IgG antibodies [10,11,12,15,28,29,30], but rare case reports have also described false serology for hepatitis A, cytomegalovirus, syphilis, and toxoplasma [13,14,15,30]. A recent study by Hanson et al. assessed the impact of IVIG therapy on a variety of common viral, bacterial, fungal, and parasitic serologies by prospectively evaluating serologic changes pre- and post-IVIG infusion in seven participants. Following IVIG administration, serologies turned positive in some of the patients for Epstein–Barr virus early D antigen, herpes simplex virus, West Nile virus, cytomegalovirus, and the endemic mycoses Histoplasma and Coccidioides. Anti-*B. henselae* IgG antibodies were not detected after IVIG administration in any of the seven patients [16]. We also prospectively studied eight patients treated with IVIG preparations of various manufacturers by performing *Bartonella* EIA prior to and after IVIG therapy. We detected transient seropositiviy following IVIG in only one patient, although increase in antibody to titers interpreted as intermediate results was identified in another four patients, while three patients remained seronegative. Our findings together with those of Hanson et al. suggest that false seropositivity after IVIG therapy may not occur consistently in all patients and may be influenced by various factors. One may hypothesize that IVIG-associated false seropositivity occurs more commonly in children, as all of our eight patients with false serology were children 1–14 years-old (Table 1), while six of the eight patients studied prospectively (Table 2) as well as all seven patients described by Hanson et al. were adults. More studies are needed to confirm this assumption. Other factors with potential impact on seropositivity may include the types of serological assays utilized for antibody testing, host-dependent response, or the immunoglobulin composition of different brands or lot-to-lot variability of various commercial IVIG preparations. A recent study observed a higher number of newly positive serologies in subjects receiving cumulative doses of IVIG in excess of 100 g and suggested a dose-response to new positive results [16].

Manufacturing of IVIG generally involves pooling of several thousand healthy plasma donors, which, after removal of the majority of IgM and IgA fractions, contains mainly IgG immunoglobulins. IVIG represents, therefore, a wide range of IgG, and is expected to include antibodies to common infectious agents that may be passively acquired, but less so to relatively rare infections such as CSD. In fact, a study performed in our laboratory has demonstrated that seroprevalence of anti-*B. henselae* antibodies in Israel is quite low as determined by finding only four (1.8%) anti-*B. henselae* IgG-positive individuals among 220 blood donors. This finding suggests that false seropositivity may not be necessarily a result of high anti-*Bartonella* antibody titers in the IVIG product, but that non-specific interaction between the donor IgG proteins and the test reagents may occur, as suggested in a recent study [30].

The misinterpretation of serologic testing is particularly important for infectious diseases for which serology is the principle diagnostic modality, and especially when diagnosis is based on IgG antibody detection rather than IgM, as in the case of CSD.

Our observations also suggest that physicians are not aware enough of the possibility of transient serologic effects of IVIG therapy, particularly when a rare pathogen is involved.

To conclude, we have demonstrated that treatment with IVIG can lead to false-positive serology for *B. henselae.* Diagnostic error is likely to occur in particular when there is a history of cat contact and a clinical presentation suggestive of CSD. Increased awareness to the misleading impact of IVIG is warranted to avoid serological misinterpretation. Repeat testing can distinguish between true and false serology. Preserving serum samples prior to administration of IVIG is currently not done routinely in clinical practice, although could be a simple procedure to follow for later clarification, if necessary. More studies are needed to understand the mechanism of IVIG false positivity.

## 4. Materials and Methods

Study design. A surveillance study of CSD has been conducted in Israel since 1991. Serology, PCR assays, and cultures for the detection of *B. henselae* infection are performed in a single laboratory, thus allowing identification of essentially all laboratory-confirmed cases of CSD in Israel. Data are recorded in the CSD Surveillance Database. A case of CSD is defined as a patient meeting the following criteria: (i) Symptoms and signs consistent with either typical or atypical CSD, in the absence of another diagnosis. (ii) One or more confirmatory laboratory results: positive serology for anti-*B. henselae* antibodies (IgM and/or IgG), positive PCR for *B. henselae* DNA, and/or positive *B. henselae* cultures. For the purpose of this study, we identified in the CSD Surveillance Database of years 1996–2019 eight patients suspected of having CSD based on clinical presentation and positive serology, which was later found to be falsely positive due to IVIG given one to six days prior to serological testing. As a control group, we have identified in our database 24 CSD patients, each with two serum samples taken 15 to <80 days apart and compared the IgG titers between the early and late serum samples. To further study this observation, we have prospectively evaluated a convenience sample of eight patients treated with IVIG for various conditions and tested serum samples taken one day prior to IVIG administration, one to three days and one to two months after IVIG administration for anti-*B. henselae* antibodies. The study was approved by the institutional review board of the Tel Aviv Sourasky Medical Center.

Laboratory tests. EIA for detection of anti-*B. henselae* antibodies was performed and interpreted as reported previously [19,20,31]. The EIA antigen is prepared as sarcosyl-insoluble, outer-membrane protein extracts of the agar-derived *B henselae* 87–66 strain (ATCC no. 49793). Sera were initially screened at 1:100 dilution and positive sera at this dilution, as determined previously [19], were tested at incremental dilutions to determine their endpoint. For the purpose of this study, we also present results expressed as arbitrary ELISA units (EU), where ≥60 EU, 50–59 EU, and <50 EU were considered positive, intermediate, and negative results, respectively. Serum samples were tested in triplicate and samples from the same patient were compared in parallel in the same ELISA plate. PCR was performed on lymph node tissue or pus aspirates using previously reported protocols [32,33].

## Figures and Tables

**Figure 1 pathogens-11-00177-f001:**
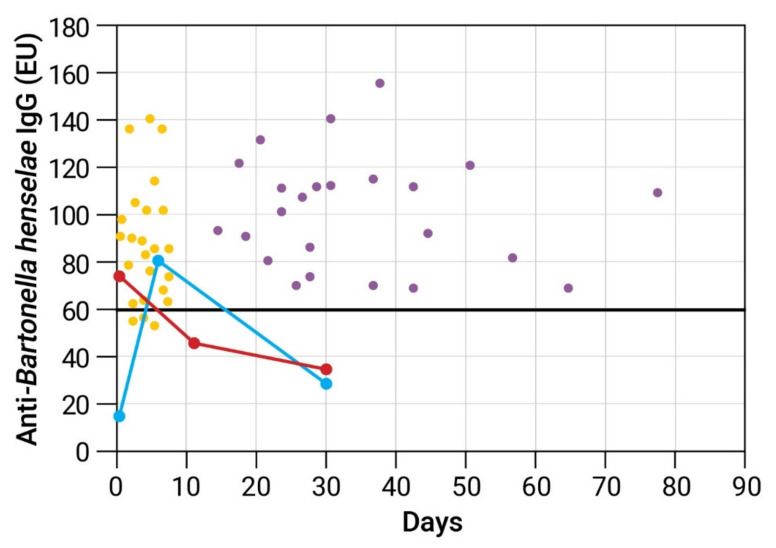
Kinetics of anti-*B. henselae* IgG antibodies determined by enzyme immunoassay in 2 representative cases with false-positive cat scratch disease (CSD) serology and in a control group of 24 CSD patients. Serum specimens were diluted 1:100. Anti-*B. henselae* IgG expressed in ELISA units (EU). Bold solid horizontal line represents cutoff point for a positive test results (60 EU). **False-positive CSD serology**. **Blue circles**—A patient with encephalitis. First serum (day 0) was tested negative 2 days prior to IVIG administration with sharp increase of IgG titers on day 6. A third specimens taken on day 30 tested negative. **Red circles**—A patient with atypical Kawasaki disease considered initially to have typical CSD. First serum specimen was taken 1 day after IVIG administration and was tested positive. Serum specimens from days 12 and 30 tested negative. **Control group** (scatter plot). Each dot represents one serum sample of a CSD patient tested for anti-*B. henselae* IgG. Each patient had 2 serum samples obtained 15–78 days apart. **Yellow dots** represent first serum samples. **Lilac dots** represent second serum samples.

**Table 1 pathogens-11-00177-t001:** Patients suspected to have cat scratch disease with false-positive anti-*Bartonella henselae* serology following intravenous immunoglobulins (IVIG) treatment ^a^.

Age-yrs/Sex	Animal Contact	Diagnosis	Clinical Manifestations and Outcome	Anti-*B. henselae* IgG after IVIG ^d^		
				1–6 Days	7–14 Days	15–30 Days
8/F ^b^	Kitten	Meningoencephalitis, possibly due to *Mycoplasma pneumoniae*	Fever, pneumonia, stupor, irritability, nuchal rigidity, progression to coma. Outcome: sever neurological damage.	Positive 1:100	Negative	Not done
4/M	None	Encephalitis	Status epilepticus, fever. Outcome: difficult-to-treat recurrent seizures	Positive ^e^ 1:200	Not done	Negative
1/F	Dog	Encephalitis	Status epilepticus, fever. Outcome: severe neurological damage	Positive 1:100	Not done	Negative
4/F	Cat	Acute transverse myelitis	Rapidly progressive paraparesis, bladder and bowel dysfunction, MRI: C5–T1; T3–T8 spinal cord thickening with high intensity signals. Outcome: recovery	Positive 1:100	Not done	Negative
13/F	None	Acute transverse myelitis	Rapidly progressive flaccid quadriparesis, Bladder and bowel dysfunction. MRI: C4–T2 high intensity signals, sensory level at T10. Outcome: transfer to rehabilitation center.	Positive 1:100	Not done	Negative
14/M	None	Guillain-Barré syndrome	Ascending weakness of 4 limbs and respiratory muscles. Outcome: transfer to rehabilitation center.	Positive 1:200	Positive 1:100	Intermediate <1:100
4/M ^c^	Kitten	Atypical Kawasaki disease	Fever, conjunctivitis, cervical lymphadenopathy, rash, anterior uveitis. Outcome: recovery	Positive 1:100	Negative	Negative
8/M	No data	Atypical Kawasaki disease	Fever, rash, conjunctivitis, arthralgia, swollen hands. Outcome: recovery	Positive 1:100	Negative	Not done

^a^ IVIG preparations of various manufacturers were given at a dose of 2 gr/kg in two divided doses; ^b^ Case 1 presented in results; ^c^ Case 2 presented in results; ^d^ Sera were initially screened at 1:100 dilution and positive sera at this dilution were tested at incremental dilutions to determine their endpoint. EIA results, expressed in arbitrary ELISA units (EU), were interpreted as follows: ≥60 EU, positive; 50–59 EU, intermediate; <50 EU, negative; ^e^
*Bartonella* serology was negative 2 days prior to IVIG administration; MRI, magnetic resonance imaging.

**Table 2 pathogens-11-00177-t002:** Prospective evaluation of patients who received intravenous immunoglobulins (IVIG) for various indications.

Age-yrs/Sex	IVIG Total Dose ^a^	Indication for IVIG	Anti-*B. henselae* IgG		
			1 Day before IVIG	1–3 Days after IVIG	1–2 Month after IVIG
46/F	2 gr/kg	POEMS syndrome	Negative	Positive 1:100	Negative ^b^
27/F	2 gr/kg	CMV pneumonia	Negative	Intermediate	Negative
55/F	2 gr/kg	ITP	Negative	Intermediate	Negative
92/F	1.5 gr/kg	ITP	Negative	Intermediate	Negative
59/F	1 gr/kg	ITP	Negative	Intermediate	Negative
9/M	2 gr/kg	SSPE	Negative	Negative	Negative
8/M	2 gr/kg	ADEM	Negative	Negative	Negative
49/F	2 gr/kg	Hypogamma-globulinemia secondary to CLL	Negative	Negative	Not done

^a^ IVIG given at 2–3 divided doses; ^b^ Serum sample was taken 2 weeks after IVIG administration; POEMS, Polyneuropathy, Organomegaly, Endocrinopathy, Monoclonal protein, Skin changes; CMV, cytomegalovirus; ITP, Idiopathic thrombocytopenic purpura; SSPE, subacute sclerosing panencephalitis; ADEM, acute disseminated encephalomyelitis; CLL, chronic lymphocytic leukemia.

## Data Availability

Not applicable.

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
