# Peer review of "Misleading Positive Serology for Cat Scratch Disease following Administration of Intravenous Immunoglobulin"

_pathogens, 2022, doi:10.3390/pathogens11020177_

Round 1

Reviewer 1 Report

General comments

This paper highlights the possible CSD overdiagnosis due to exogenous antibodies and the importance of repeated serological test for Bartonella infection in patients-recipients of IGIV. All in all, it is an interesting and well written manuscript.  

Specific comments:

LL 52 The Bartonella culture is not only demanding but also time-consuming and should be included in the disadvantages.

LL 126 Could you present the ELISA results in the same table (Table 1)?

LL 141 The figure 1 should provide the whole picture and contain the false positives and the controls e.g., as a scatter plot of the two groups marked with different colors or otherwise. The representative as selected by the authors may introduce bias and do not present all the information.

LL 193-210 Please provide a comparison of specificity/sensitivity with B. henselae IFA IgG which is the most widely used and considered by many as the gold standard. If there is any other report from other laboratory using this EIA should be provided.

LL 236-238 A higher frequency of false seropositivity especially in children is hypothesized and is not confirmed by the low numbers of the participants in the study. Should be rephrased accordingly.

LL 220 – 259 If there are data on how frequently IGIV is administered and in what health settings eg., pediatric wards, or other related info, would be good to report somewhere in the discussion as it helps assess the magnitude of potential CSD overdiagnosis.

LL 246-256 if the authors are aware of any standard -universal - screening protocol for donors for IGIV regarding infectious diseases or if non-existent, should report it.

LL 272-289 Please provide a time frame that the study took place. You describe that a CSD surveillance is in place since 1991 in Israel but other than that I do not see any other description regarding time. Also provide some brief description regarding the place where the patients, controls etc. were presented.  

Reviewer 2 Report

An interesting paper and well written. Just a few suggestions for improvement:

  1. For cases 1 and 2, that are reported in detail, I would recommend reporting all the diagnostic tests that were done and were negative. This data will be of interest to readers who will be thinking about the possible correct diagnosis of these 2 patients. A "workup" involving "a host of" assays is not really sufficient information in my opinion. Readers deserve to know the assays done and the results.
  2. line 134. "...titer increase from 86+/-26 EU in the first serum sample to 99+/-25 EU in the second sample...". This is clearly a stationary titer, NOT an increasing titer, due to errors associated with each reading.
  3. in Table 2 the first case (46/F) should be removed as there is no serum titer reported at 1-2 months after IVIG, so there is no way of knowing if the this patient has indeed become sero-negative.
  4. The "Author Contributions" section needs to be completed.
